# Isolation and Comparative Study on the Characterization of Guanidine Hydrochloride Soluble Collagen and Pepsin Soluble Collagen from the Body of Surf Clam Shell (*Coelomactra antiquata*)

**DOI:** 10.3390/foods8010011

**Published:** 2019-01-01

**Authors:** Jiulin Wu, Xiaoban Guo, Hui Liu, Li Chen

**Affiliations:** Institute of Biomedical and Pharmaceutical Technology, Fuzhou University, Fuzhou 350002, China; gxb_0408@163.com (X.G.); worldliuhui@163.com (H.L.)

**Keywords:** surf clam shell, guanidine hydrochloride soluble collagen, pepsin soluble collagen, isolation, characterization

## Abstract

The aim of this study was to characterize the collagens from the body of surf clam shell (*Coelomactra antiquata*). Guanidine hydrochloride and pepsin were used to extract collagens. Guanidine hydrochloride soluble collagen (GSC) and pepsin soluble collagen (PSC) were separately isolated from the body of surf clam shell. Results showed that the moisture, protein, carbohydrate, and ash contents of the body of surf clam shell were 82.46%, 11.56%, 3.05%, and 2.38%, respectively, but the fat content was only 0.55%. The yields were 0.59% for GSC and 3.78% for PSC. Both GSC and PSC were composed of α_1_ and α_2_ chains and a β chain, however, GSC and PSC showed distinct differences from each other and the type I collagen from grass carp muscle on sodium dodecyl sulfate-polyacrylamide gel electrophoresis (SDS-PAGE). GSC and PSC contained glycine as the major amino acid and had imino acid of 150 and 155 residues/1000 residues, respectively. Fourier transform infrared spectroscopy (FTIR) spectra of GSC and PSC revealed the presence of a triple helix. The GSC appeared to have a dense sheet-like film linked by random-coiled filaments and PSC had fine globular filaments under scanning electron microscopy (SEM). The maximum transition temperature (T_max_) of GSC and PSC was 33.05 °C and 31.33 °C, respectively. These results provide valuable scientific information for the texture study and development of surf clam shell or other bivalve mollusks.

## 1. Introduction

Marine-based species comprise approximately one half of the total global biodiversity, and the oceans and aquatic environments in general offer plenty of resources for novel bioactive components. Marine species contain bioactive compounds and much attention has been paid to them, as they play pivotal roles in disease prevention and the maintenance of human health. These marine bioactive compounds exhibit significant biological properties that contribute to their nutraceutical and pharmaceutical potential and are also considered to be safer alternatives to some existing synthetic drugs [1,2,3]. Proteins (including collagen) or peptides are very important bioactive compounds that can be used in food and medicine [4,5].

Collagens, the major component of extracellular matrix (ECM) proteins, are a heterogeneous family of structural proteins representing nearly one-third of the total proteins. So far, approximately 29 types of collagens numbered as I to XXIX have been identified, and each type has considerably different molecular structures, amino acid sequences, and functions [6,7,8]. Most of these collagens have potentially widely commercial applications in food, cosmetic, biomedical, and pharmaceutical industries [9,10]. Gelatin, which is a denatured form of collagen, is utilized in confections, low fat spreads, dairy, baked goods, and meat products [11]. Collagen-based biomaterials, such as scaffolds and films, have been applied in medical research, as well as tissue engineering, such as drug delivery, soft tissue augmentation, blood valve prosthesis, and wound dressings [11,12]. Collagen and gelatin are usually hydrolyzed for the production of bioactive peptides, which have tremendous potential to be utilized in nutraceuticals and pharmaceuticals [3]. While muscle collagen is responsible for the integrity of myocommata and the mechanical properties of muscle, which is very important in the development of the texture of raw or cooked meat from aquatic animals [13].

Traditionally, collagen has been extracted from bovine and porcine skin and bones. However, collagens obtained from porcine skin or bone are prohibited for some religious and ethnic groups, such as Jews and Muslims [14,15]. Additionally, the outbreak of bovine spongiform encephalopathy (BSE), transmissible spongiform encephalopathy (TSE), and foot and mouth disease (FMD) have resulted in anxiety among users of collagen and collagen-derived products from these land animals and thus further limit the application of collagen [9]. As a consequence, alternative sources of collagen, especially from aquatic animals, have received increasing attention [16]. There are immense untapped marine sources that could be utilized in order to obtain different types of collagen. Marine collagens have been reported to be biomaterials that are potentially usable as alternatives to conventional mammalian collagens [9]. There have been a large number of papers describing collagens from aquatic vertebrates, which have been characterized in detail [6,8,11,15,17,18,19,20,21]. However, our knowledge on aquatic invertebrate collagen still remains scanty [9].

Bivalve mollusks, which belong to commercially important groups of aquatic invertebrate animals, are used as a food resource worldwide. As for collagen in bivalve mollusks, attempts have been made to extract and characterize the physicochemical and biochemical properties of collagens from only several mollusks, such as mussel (*Geukensia demissa*) [22], pearl oyster (*Pinctada fucata*) [23], and hard clam (*Meretrix lusoria*) [24]. It has been found that the physical and chemical properties of these collagens are obviously different from those of fish species and land animals, which are more likely to be used as suitable biomaterials in commercial applications. Further, collagens in the muscles of marine animals play a key role in maintenance of meat texture and perform functions different from that of their vertebrate counterparts [25,26].

Surf clam shell (*Coelomactra antiquata*) is one of the most important marine bivalve mollusk species in China, especially in the east and southeast of China. Nevertheless, no information regarding the characteristics of collagens from surf clam shell has been reported so far. Therefore, the aim of this study was to isolate and characterize collagens from the body of surf clam shell. The results will hold significance to our understanding of collagen polymorphism in marine invertebrates in relation to molecular evolution and structure–function relationships.

## 2. Materials and Methods

### 2.1. Raw Materials

Live surf clam shells (*Coelomactra antiquata*) were obtained from Xiyingli seafood market in Fuzhou, Fujian Province. The shell was removed manually. Muscle tissues (mantle and adductor) were taken out and washed with chilled distilled water three times. The samples were collected and immediately used for the experiments.

### 2.2. Chemicals Reagents

All reagents were of analytical grade. Sodium dodecyl sulfate (SDS), Coomassie Blue R-250, and N,N,N′,N′-tetramethylethylenediamine (TEMED) were procured from Sigma-Aldrich Chemical Company (St. Louis, MO, USA). Molecular weight markers were obtained from Fermentas (Burlington, CA, USA). Other chemicals and reagents were purchased from Sinopharm Chemical Reagent Co. (Shanghai, China). Type I collagen from the skeletal muscle of grass carp was prepared in our own laboratory.

### 2.3. Extraction of Guanidine Hydrochloride Soluble Collagen (GSC)

GSC was extracted following the method of Mizuta et al. (1997) with slight modifications [27]. To remove non-collagenous proteins, muscle tissues were homogenized in ten volumes (*v*/*w*) of 0.1 M NaOH, and extracted for 24 h with gentle stirring at 4 °C. The residue after alkali extraction (RS-AL) was washed thoroughly with distilled water, and extracted with twenty volumes (*v*/*w*) of 50 mM Tris–HCl, pH 7.0, containing 4 M guanidine hydrochloride (G/HCl) for 24 h at 4 °C. The supernatant was collected by centrifugation at 15,000× *g*, at 4 °C for 30 min, dialyzed against distilled water overnight and then against 0.5 M acetic acid containing 2.0 M NaCl at 4 °C. After centrifugation at 15,000× *g* for 20 min, the resultant precipitate was collected, dialyzed against distilled water at 4 °C for 48 h, and lyophilized. The preparation obtained was referred to as G/HCl-soluble collagen (GSC).

### 2.4. Extraction of Pepsin Soluble Collagen (PSC)

The insoluble remaining matter after the G/HCl extraction of the RS-AL was washed thoroughly with distilled water and then digested with porcine pepsin in 0.5 M acetic acid at an enzyme/substrate ratio of 1:20 (*w*/*w*) for 48 h at 4 °C. After centrifugation at 15,000× *g* for 20 min, at 4 °C, the supernatant was obtained. NaCl powder was slowly added until the final concentration of 0.6 M was reached. It was then left to stand at 4 °C for 12 h and then centrifuged at 15,000× *g* for 20 min at 4 °C. The resulting precipitate was dissolved in 0.5 M acetic acid and dialyzed against 0.02 M Na_2_HPO_4_ at 4 °C for 48 h. The dialysate was centrifuged at 15,000× *g* for 20 min at 4 °C. The resulting precipitate was dissolved in 0.5 M acetic acid and dialyzed against distilled water at 4 °C for 48 h. The resulting dialysates was freeze-dried and considered as pepsin soluble collagen (PSC).

### 2.5. Biochemical Compositions Analysis and the Collagen Yield of Surf Clam Shell

The moisture, protein, and ash contents of surf clam shell were determined according to the method of Meng et al. (2007) [28]. The yields of GSC and PSC were calculated based on the dry weight of starting materials. All the experiments were replicated three times.
Yield (%) = [Weight of lyophilized collagen (g)/Weight of initial dry whole soft body (g)] × 100(1)

### 2.6. Sodium Dodecyl Sulfate–Polyacrylamide Gel Electrophoresis (SDS–PAGE)

SDS–PAGE of collagens from muscle tissues was carried out according to the method of Wu et al. (2014) with slight modifications [29]. Solubilized samples were mixed with one quarter of SDS sample buffer (200 mM Tris–HCl, pH 6.8, containing 8% SDS (*w*/*v*), 0.4% bromophenol blue, and 40% (*v*/*v*) glycerol), then homogenized and heated in a water bath for 5 min. Samples (12 μg protein/each sample) were loaded onto the polyacrylamide gel (8%), respectively and then electrophoresed at a constant current of 12 mA. After electrophoresis, the gels were stained with 0.1% (*w*/*v*) Coomassie Blue R-250 in 50% (*v*/*v*) methanol and 7.5% (*v*/*v*) acetic acid at 70 °C for 30 min. Finally, the gels were destained with a mixture of 30% (*v*/*v*) methanol and 10% (*v*/*v*) acetic acid for 30 min twice by gently shaking. The protein molecular weights were calculated based on the unstained protein molecular weight markers (Ferments, Burlington, CA, USA).

### 2.7. Amino Acid Analysis

Amino acid analysis of the GSC and PSC from surf clam shell was performed according to the method of Lin et al. (2017) with slight modifications [30]. GSC and PSC were hydrolyzed respectively in 6 M hydrochloric acid at 110 °C for 24 h in the absence of oxygen. The hydrolysates were analyzed on a Hitachi L-8800 auto amino acid analyzer (Hitachi, Tokyo, Japan) with a mobile phase flow of 0.400 mL min^−1^ and the flow of ninhydrin solution was set at 0.350 mL min^−1^. The content of amino acid was expressed as residues/1000 residues.

### 2.8. Analysis of Fourier Transform Infrared Spectroscopy (FTIR)

Amide band patterning of GSC and PSC was analyzed using a Nicolet AVATAR 360 FTIR spectrometer (Nicolet Co., Madison, WI, USA) according to the method of Lin et al. (2017) with slight modifications [30]. Under drying condition, about 5 mg lyophilized collagen sample and 100 mg potassium bromide (KBr) were ground together using a mortar and pestle. The sample was subjected to a pressure of about 5 × 10^6^ Pa in an evacuated die to produce a 13 × 1 mm clear transparent disk. The spectrum was obtained with 32 scans per sample ranging from 4000 to 400 cm^−1^. The resulting spectral data was analyzed using ORIGIN 8.0 software (Thermo-Nicolet, Madison, WI, USA).

### 2.9. Scanning Electron Microscopy (SEM)

The morphological characteristics of the pretreated collagens were studied by SEM (Nova NanoSEM 230, Hillsboro, OR, USA) according to the method of Tziveleka et al. (2017) with slight modifications [7]. The samples were mounted on stubs, sputter-coated with gold, and then observed for surface morphology at various magnifications. The SEM observations were made at 15 kV accelerating voltage. A higher vacuum (HV) mode and secondary electron image (SEI) were employed to scan the microscopic images of collagen matrix.

### 2.10. Determination of Denaturation Temperature

GSC and PSC were separately rehydrated by adding deionized water at a solid to solution ratio of 1:40 (*w*/*v*). The mixtures were allowed to stand for 2 days at 4 °C prior to analysis. Differential scanning calorimetry (DSC) was performed using a differential scanning calorimeter model DSC 7 (Perkin Elmer, Norwalk, CT, USA) according to the method of Yang et al. (2016) with slight modifications [31]. Calibration was run using the Indium thermogram. The samples (5–10 mg) were accurately weighed into aluminum pans and sealed. The samples were scanned at 1 °C min^−1^ over the range of 25–50 °C using iced water as the cooling medium. An empty pan was used as the reference. The maximum transition temperature (T_max_) was estimated from the thermogram. The total denaturation enthalpy (ΔH) was estimated by measuring the area of the DSC thermogram.

## 3. Results and Discussion

### 3.1. Biochemical Compositions

Surf clam shell (*Coelomactra antiquata*) contained very high moisture content, which was about 82.46 ± 0.12%. The protein, carbohydrate, and ash contents of surf clam shell, which were 11.56 ± 0.08%, 3.05 ± 0.05% and 2.38 ± 0.05%, respectively, were much lower than the moisture content. It is noteworthy that the fat content was very low (only 0.55 ± 0.09%). Tabakaeva et al. (2018) also reported that the two commercially significant edible bivalve mollusk species (*Anadara broughtonii* and *Mactra chinensis*) had low fat content, and that protein and carbohydrates were their main components [32].

### 3.2. The Yields of GSC and PSC from Surf Clam Shell

The approximate collagen content of surf clam shell was 4.37 ± 0.04% (on a dry weight basis) of protein for the whole soft body, which was similar to the yields of collagen from *Mytilus galloprovincialis* and *Septifer virgatus* [24]. When the GSC and PSC were extracted from the whole soft body of surf clam shell, the respective yields of 0.59 ± 0.03% and 3.78 ± 0.04% (on a dry weight basis) were obtained. So far, few GSC yields have been previously reported. Based on the lyophilized dry weight, the PSC yield of rhizostomous jellyfish (*Rhopilema asamushi*) was as high as 35.2% [33], whereas the PSC yield of silver carp scales was only 2.32% [34]. Therefore, the collagen yields might be associated with species and preparation methods. Also, seasonal change might also influence the yield of collagen. Olaechea et al. discovered that the collagen content of the foot muscle of the disk abalone (*Haliotis discus*) showed seasonal change that corresponded well with the meat toughness [35].

### 3.3. Protein Patterns

Electrophoretic patterns of GSC and PSC analyzed by SDS-PAGE under non-reducing condition are shown in Figure 1, along with type I collagen from the muscle of grass carp. It revealed that both of the two collagens comprised at least two different α chains (α_1_ and α_2_) and high-molecular-weight components including β chain. Both GSC and PSC also contained γ chain-sized components to some extent. However, there were considerable differences in the SDS–PAGE patterns, suggesting minor differences in the structure of GSC and PSC. The GSC showed two α chain-sized components (α_1_ and α_2_), with molecular weights estimated to be in the range of 125–135 kDa, and one β chain-sized component with slower mobility than those of the corresponding components in grass carp type I collagen. Compared to GSC, the PSC showed mainly two α chain-sized components (α_1_ and α_2_) with much lower molecular weight components and a β chain-sized component with higher molecular weight. Moreover, the molecular weights of both α chain-sized components and the β chain-sized component of GSC and PSC were higher than that of type I collagen from grass carp muscle. Further, the density of the β chain of GSC was higher than that of PSC, which might indicate that GSC has more intramolecular or intermolecular cross-links than that in PSC. On the contrary, the relative staining intensity of the α_1_ and α_2_ chains was apparently higher in PSC than that in GSC. This was explained by conversions of some β or γ chain-sized component in the PSC matrix to α-components by the treatment with pepsin. Pepsin cleaves the crosslink containing telopeptide, and the β-chain is converted to two α-chains [36]. 

The GSC and PSC extracted from pearl oyster (*Pinctada fucata*) and *Crassostrea gigas* exhibited quite a similar pattern [23,24]. However, in our current experiment, the results showed that the SDS–PAGE patterns of the GSC and PSC were considerably different, which was most likely due to the seasons, species, and preparation methods. Also, they were very different from that of collagen from Australasian Snapper (*Pagrus auratus*) [11], seabass (*Lates calcarifer*) [20], squid (*Todarodes pacificus*) [27], and soft-shelled turtle calipash [31]. 

### 3.4. Amino Acid Composition

The amino acid compositions of both GSC and PSC from surf clam shell, expressed as residues per 1000 total residues, are shown in Table 1. GSC and PSC were rich in glycine (244 and 254 residues/1000 residues, respectively), which was the major amino acid in collagen. The glycine contents of GSC and PSC were lower than that of PSC from bighead carp (*Hypophthalmichthys nobilis*) [6], unicorn leatherjacket (*Aluterus monocerous*) [14], silver carp (*Hypophthalmichthys molitrix*) [34], and marine crab (*Scylla serrate*) [26]. Generally, glycine in collagens represents nearly one-third of the total residues and occurs at every third residue in collagens, except for the first 14 amino acid residues from the N-terminus and the first 10 residues from the C-terminus [37]. Glutamic acid contents of GSC and PSC were 118 and 119 residues/1000 residues respectively, thus representing the second most abundant amino acid. Low contents of histidine were found to be 11 and 9 residues/1000 residues in GSC and PSC, respectively. The cysteine contents of GSC and PSC were only 1 and 2 residues/1000 residues, respectively.

The imino acid (proline and hydroxyproline) were unique amino acids found in collagen. Their contents in GSC and PSC were 150 and 155 residues/1000 residues, respectively. However, so far, little information on the imino acid of GSC from bivalve mollusks has been reported. The imino acid content of PSC from surf clam shell was higher than those of the PSC from the fresh body wall of *A. leucoprocta* (141 residues/1000 residues) [30], and collagen from blue shark cartilage (122 residues/1000 residues) [38], but lower than those of the PSC from the skin of bighead carp (165 residues/1000 residues) [6] and swim bladders of yellowfin tuna (169 residues/1000 residues) [39]. Imino acids contribute to the stability of the helix structure of collagens [40]. In addition, the imino acid content has been known to determine the thermal stability of collagen and the formation of junction zones via hydrogen bondings. Hydroxyproline plays a significant role in stabilizing the triple helical structure by the formation of interchain hydrogen bonds through the hydroxyl group [20]. These results suggested that the slightly different imino acid composition of GSC and PSC from surf clam shell tended to result in the distinct differences in structure and thermal stability.

Marine collagens are a promising source for producing bioactive peptides. Researchers have found some notable activities of marine collagen peptides, such as skin antiaging activity, antioxidant activity, antihypertensive activity, antimicrobial activity, anti-HIV-1, wound healing, and iron-binding [9]. Collagen peptide containing several specific amino acids presented relevant activities. Many researchers have previously found that low molecular weight collagen peptides (di- and tripeptides), especially those with C-terminal Pro or Hyp residues, have exhibited numerous bioactivities including antibacterial, antioxidative, angiotensin converting enzyme (ACE) inhibitory properties, etc. [9]. It is well documented that dietary supplementation with arginin improves glycemic control [41]. GSC and PSC from surf clam shell contained 18 amino acids. Therefore, the GSC and PSC may represent potential resources for producing bioactive peptides.

### 3.5. Fourier Transforms Infrared (FTIR) Spectra of Collagen

FTIR spectra of GSC and PSC exhibited the characteristic peaks of amide A and B as well as amide I, II, and III bands (Figure 2). FTIR spectra for the GSC and PSC were similar to other collagens [6,39]. The amide A band is associated with the N–H stretching frequency. The amide A band positions of GSC and PSC were found at 3416 and 3374 cm^−1^, respectively, which are shown in Figure 2b. The absorption characteristic of amide A is commonly associated with the N–H stretching band and shows the existence of hydrogen bonds. A free N–H stretching vibration occurs in the range of 3400–3440 cm^−1^, and when the N–H group of a peptide is involved in hydrogen bonding, the position is shifted to lower frequencies, usually 3300 cm^−1^ [42]. The result indicated that more N–H groups of GSC were involved in hydrogen bonding than those of PSC. Amide B bands of GSC and PSC were observed at 2925 and 2921 cm^−1^, respectively, which represent the asymmetrical stretch of CH_2_ [43]. 

The amide I band positions of GSC and PSC were observed at 1654 and 1667 cm^−1^, respectively. The amide I band, with characteristic frequencies in the range from 1600 to 1700 cm^−1^, is mainly associated with the stretching vibrations of the carbonyl group (C=O bond) along the polypeptide backbone [38]. Amide II is generally responsible for the combination of the NH in plane bend and the CN stretching vibration [38]. The amide II band of GSC (1540 cm^−1^) was found at a lower wavenumber compared to that of PSC (1556 cm^−1^), suggesting that there was a stronger hydrogen bond in GSC. The absorption between the 1236 and 1452 cm^−1^ bands (amide III) demonstrated the presence of a helical structure and also suggested the helical arrangement of the two collagens, with the GSC absorption ratio at 1234 cm^−1^ and the PSC absorption ratio at 1242 cm^−1^ [30]. Additionally, the absorption peaks around 1451–1462 cm were also found in GSC and PSC. This corresponded well to the pyrrolidine ring vibration of proline and hydroxyproline, as previously described [24].

The FTIR investigations of GSC and PSC from the surf clam shell had some slight differences, which indicated discrepancies in the secondary structures of the collagens. The FTIR spectra were consistent with the results of SDS–PAGE.

### 3.6. Morphological Characterization of Collagens

The morphological structures of the extracted collagens (GSC and PSC) were observed under SEM micro-photography with higher magnification (Figure 3). All the collagens looked like soft white sponge with loose and porous structure by the naked eye. However, the GSC appeared to be a dense irregular sheet-like film linked by random-coiled filaments under SEM, and the surface was partially wrinkled. This was possibly because of dehydration during lyophilizing (Figure 3a), which was in agreement with the skin collagen of *Amur sturgeon* [21]. Although not well organized, the intersecting sheet-like films were not parallel but entangled in individual bundles in the three-dimensional structure (Figure 3b).

Semblable porous matrix with good interconnectivity in PSC was quite different from GSC (Figure 3c), which was observed to have a complex meshwork form and contact with some fibrils. These collagen fibrils across the porous matrix varied in width and thickness, and intertwined with each other. In addition, the fibrillar appearance observed in PSC changed into an amorphous structure, characterized by a structure composed of strips (Figure 3d). The lyophilized PSC was loose and endowed with uniform and regular alveolate pores due to the evaporation of fluid; also, the pore size of the collagen sponge increased at higher water content during preparation [39]. Additionally, certain areas contained disorganized nanofibers.

Efforts have been made to discover the uses of marine collagen, including possible uses for drug delivery systems, tissue engineering, cosmetics, and nutricosmetics [9]. Uniform and regular network structures of sponges, as drug carriers are propitious, not only for well-proportioned distribution for other drugs, but also for evaporation of fluid [44]. Architectural features such as porosity, pore size, and specific surface areas are wildly considered as important factors for a biomaterial to understand their biomedical importance [45]. In addition, other architectural features, such as pore shape, pore wall morphology, and interconnectivity of collagen, have also been suggested for use in cell seeding, growth, gene expression, migration, mass transport, and new tissue formation. According to SEM, the extracted collagens form surf clam shell may be used as suitable biomaterials.

### 3.7. Thermal Transition

The maximum transition temperature (T_max_) and enthalpy (ΔH) of GSC and PSC from surf clam shell in deionized water are shown in Figure 4. The T_max_ values of GSC and PSC from surf clam shell were measured at 33.05 °C (ΔH = 0.3667 J g^−1^) and 31.33 °C (ΔH = 0.451 J g^−1^), respectively. Thermal stability of collagen was governed by the pyrrolidine rings of proline and hydroxyproline and partially by hydrogen bonding through the hydroxyl group of hydroxyproline [18]. A slightly higher denaturation enthalpy (ΔH) value was found for PSC in comparison to that of GSC. The removal of telopeptides might lead to a more ordered structure of PSC, in which higher energy was required. The presence of imino acids, particularly hydroxyproline in GSC and PSC, might contribute to the stabilization of the triple helix structure through hydrogen bonding in coil-coiled α-chains [17]. The T_max_ values of GSC and PSC were higher than those of collagens from carp scales (about 28 °C) and the mesoglea collagen from rhizostomous jellyfish (*R. asamushi*) (about 28.8 °C) [33], which suggested a higher thermal stability of the former examples. Interestingly, the T_max_ of PSC from surf clam shell collagens were discovered to be slightly lower than those of collagen from silver carp scales (35.5 °C) [34]. The thermal stability of collagens not only depend on the imino acid content but also directly correlate with the environmental and body temperatures of fish species [46]. Thus, the thermal properties of GSC and PSC were influenced by the tissues used for collagen extraction compared to mammalian gelatin.

## 4. Conclusions

Two different collagens of GSC and PSC were successfully isolated from the surf clam shells. Our results showed that the GSC and PSC had slight differences in molecular weights, amino acid composition, morphological structures, and thermal stability, which implied that GSC and PSC maybe two different kinds of collagens. The properties of GSC and PSC from surf clam shells also showed obvious differences compared to those of fish species. FTIR investigations showed the existence of helical arrangements of the two collagens. Therefore, guanidine hydrochloride and pepsin added extraction could serve as a tool for obtaining collagens without a marked effect on the triple-helical structure. From this study, these results could provide a valuable scientific basis for the study of the texture and development of surf clam shell or other bivalve mollusks. The results also suggest that the collagens can potentially serve as an alternative source of collagen for further application in food, pharmaceutical industries, cosmetic, suitable biomaterial, and other applications. 

## Figures and Tables

**Figure 1 foods-08-00011-f001:**
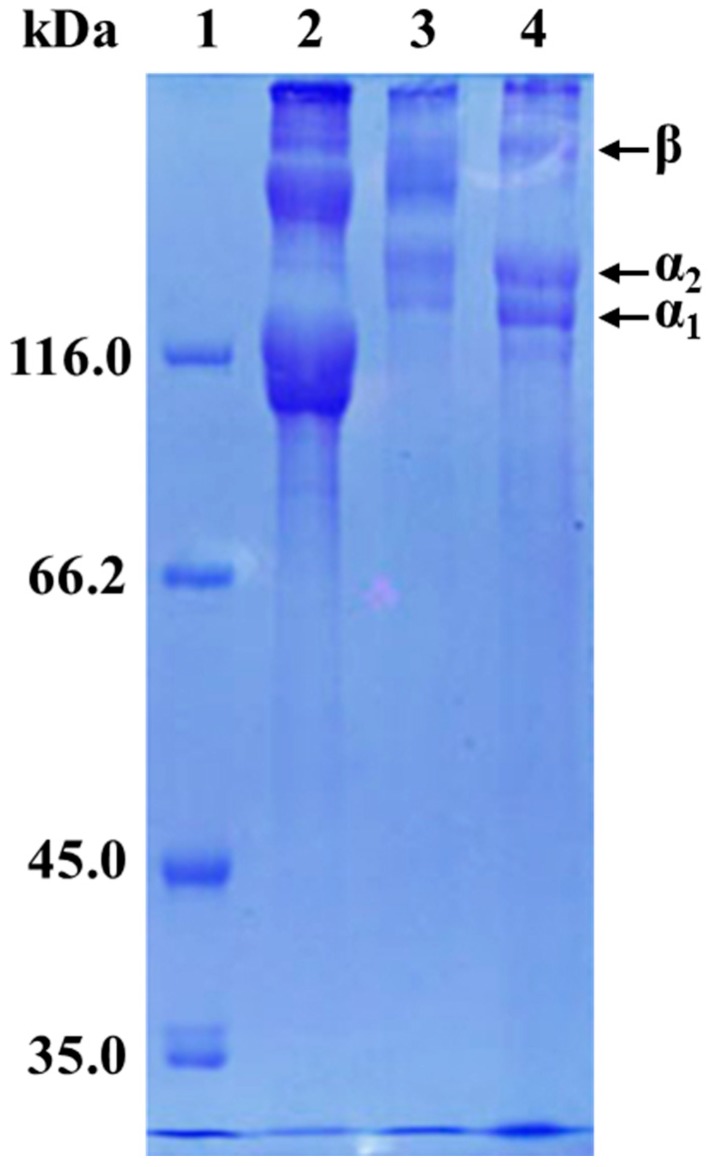
Sodium dodecyl sulfate-polyacrylamide gel electrophoresis (SDS-PAGE) patterns of collagens from surf clam shell (*Coelomactra antiquata*). Lane 1, molecular weight marker; Lane 2, Type I collagen from the skeletal muscle of grass carp; Lane 3, guanidine hydrochloride soluble collagen (GSC); Lane 4, pepsin soluble collagen (PSC).

**Figure 2 foods-08-00011-f002:**
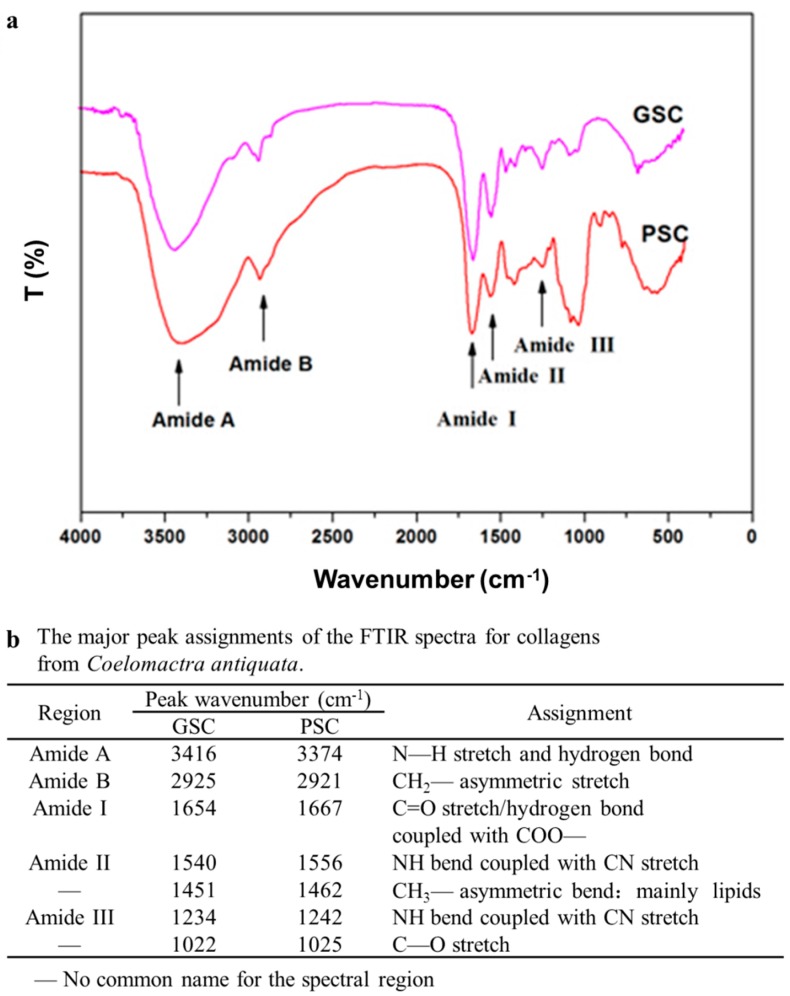
(**a**) Fourier transform infrared spectroscopy (FTIR) spectra of guanidine hydrochloride soluble collagen (GSC) and pepsin soluble collagen (PSC) from surf clam shell (*Coelomactra antiquata*); (**b**) The major peak assignments of the FTIR spectra for collagens from surf clam shell (*Coelomactra antiquata*).

**Figure 3 foods-08-00011-f003:**
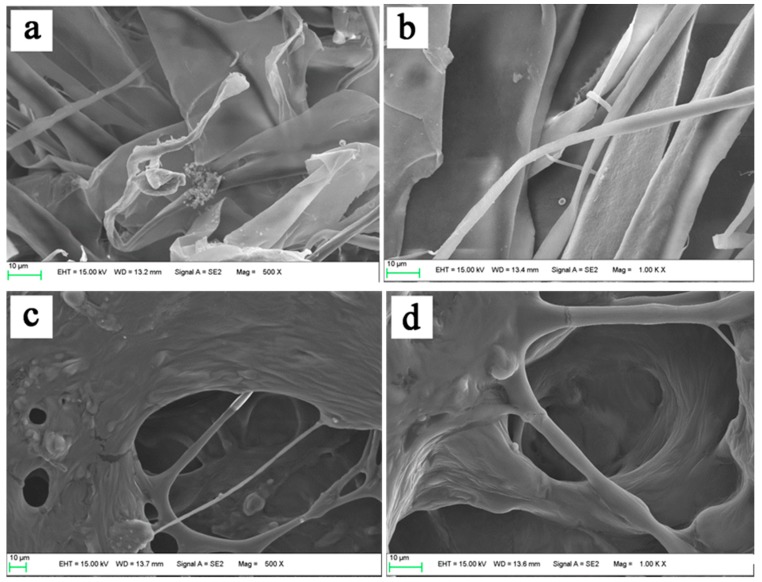
Scanning electron microscopic plates exhibiting the micro-structure of GSC (**a**,**b**) and PSC (**c**,**d**) isolated from surf clam shell (*Coelomactra antiquata*).

**Figure 4 foods-08-00011-f004:**
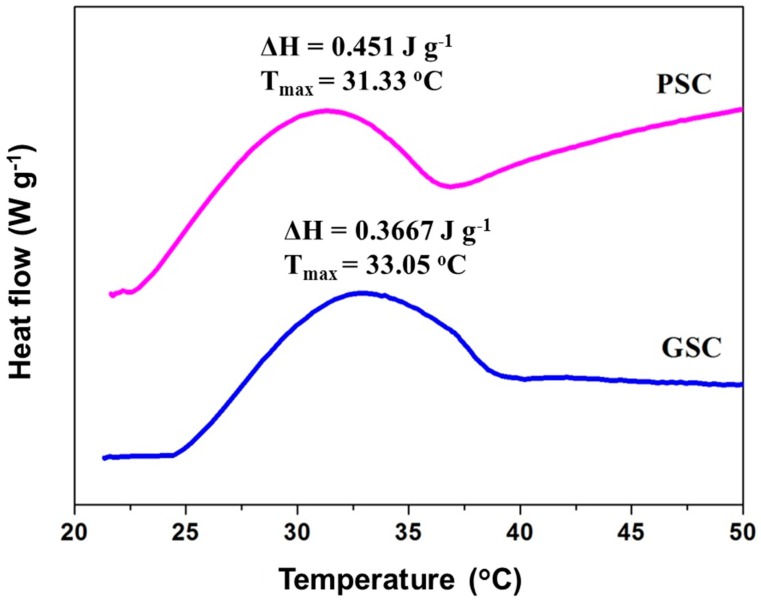
Differential scanning calorimetry (DSC) thermogram of GSC and PSC from surf clam shell (*Coelomactra antiquata*). ΔH: denaturation enthalpy; T_max_: maximum transition temperature.

**Table 1 foods-08-00011-t001:** Amino acid composition of guanidine hydrochloride soluble collagen (GSC) and pepsin soluble collagen (PSC) from *Coelomactra antiquata* (results are expressed as residues/1000 residues).

Amino Acids	GSC	PSC
Alanine	70	67
Cystine	1	2
Aspartic acid	81	76
Glutamic acid	118	119
Phenylalanine	17	16
Glycine	244	254
Histidine	11	9
Isoleucine	26	24
Lysine	35	22
Leucine	42	44
Methionine	14	15
Proline	85	86
Arginine	52	65
Serine	55	50
Threonine	35	34
Valine	32	32
Tyrosine	17	16
Hydroxyproline	65	69
Total	1000	1000
Imino acid	150	155

GSC: guanidine hydrochloride soluble collagen; PSC: pepsin soluble collagen.

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
