# Peer review of "Isolation and Comparative Study on the Characterization of Guanidine Hydrochloride Soluble Collagen and Pepsin Soluble Collagen from the Body of Surf Clam Shell (Coelomactra antiquata)"

_foods, 2019, doi:10.3390/foods8010011_

Round 1

Reviewer 1 Report

Review of Manuscript ID: foods-412483

Title: Isolation and Comparative Study on the Characterization of Guanidine Hydrochloride Soluble Collagen and Pepsin Soluble Collagen from the Body of Surf Clam Shell (Coelomactra antiquate).

Bearing in mind the risk related to the consumption of products involving animal collagen,     I believe that the subject of research undertaken by the authors of the manuscript regarding alternative sources of collagen acquisition, including aquatic animals, I consider to be important and worth continuing.

I recommend the manuscript to be published after taking into account minor comments.

L 14 “……0,59%” replace “…0,59 %”, include change throughout the text.

L82, L88, 89 Complete at what temperature the dialysis and centrifugation steps were carried outL30  fill up your keywords

L104  Complete with what molecular weight the standards were used.

L107,108  Complete the literature reference and analysis conditions on the analyzer.

L110, 117, 123 Complete the literature reference

L128, 129 – “…..1 °C/min. replace “… 1°C min-1

L271 “…..J/g replace “… J g-1

Figure 4 - “…..J/g  replace “… J g-1

References 3 and 4  - Complete the page numbers

Author Response

Thank you very much for your good comments and careful review. The responds to your comments are as following:

L 14 “……0,59%” replace “…0,59 %”, include change throughout the text.

Response: We have revised according to the comment.

L82, L88, 89 Complete at what temperature the dialysis and centrifugation steps were carried outL30 fill up your keywords

Response: The temperature was added in the dialysis and centrifugation steps. Keywords have been revised.

L104  Complete with what molecular weight the standards were used.

Response: The information of molecular weight the standards (unstained protein molecular weight markers (Ferments, Burlington, CA)) was added.

L107,108  Complete the literature reference and analysis conditions on the analyzer.

Response: The reference Lin et al (2017) and analysis conditions were added as shown in L134, 137.

L110, 117, 123 Complete the literature reference

Response: The literature references Lin et al (2017), Tziveleka et al (2017) and Yang et al (2016) were added as shown in L141, 149, 158, respectively.

L128, 129 – “…..1 °C/min. replace “… 1°C min-1

Response: It was revised as shown in L160

L271 “…..J/g replace “… J g-1

Response: It was revised as shown in L320

Figure 4 - “…..J/g  replace “… J g-1

Response: It was revised as shown in Figure 4.

References 3 and 4  - Complete the page numbers

Response: It was revised as shown in L160

Reviewer 2 Report

Dear Editor,

I am pleased to review the assigned manuscript. Overall, a proposed review looks very good to me – Authors really did a wonderful job and presented very nice & relevant literature. I have few suggestions that can be incorporated into revised version.

1.       Abstract looks good but just wondering if it can cover the whole theme

2.       The introduction should be better organized. Some of the sentences are not well structured, should be clarified and rewritten. It is advice to link the story in a better way in an introduction to convey a proper message to readers.

3.       I suggested some of the latest references, it would improve content and the novelty of the research. Authors can incorporate the related research carried out at the University of Queensland, Australia.

Marine-based species comprising approximately one half of the total universal biodiversity, the oceans and aquatic environment in general offer a plenty of resource for novel bioactive components. Marine species comprises bioactive compounds and much attention has been paid to them as they play pivotal role in disease prevention and maintenance of human health. These marine bioactive compounds exhibit significant biological properties that contribute to their nutraceutical and pharmaceutical potential and are also considered to be safer alternatives to some existing synthetic drugs. These bioactivities include anti-oxidant, anti-thrombotic, anti-coagulant, anti-inflammatory, anti-proliferative, anti-hypertensive, anti-diabetic and cardio-protection activities, making them attractive nutraceuticals and pharmaceutical compounds.

·         Marine-Based Nutraceuticals: An Innovative Trend in the Food and Supplement Industries. Marine drugs, 13(10), 6336.

·         Current and potential uses of bioactive molecules from marine processing waste. J. Sci. Food Agric., 96: 1064–1067.

·         Therapeutic potential of abalone and status of bioactive molecules: A comprehensive review. Critical reviews in food science and nutrition, 57(8), 1742-1748.

·         Marine bioactive compounds and health promoting perspectives; innovation pathways for drug discovery. Trends in Food Science & Technology, 50, 44-55

·         Marine Processing Waste - In Search of Bioactive Molecules. Nat Prod Chem Res 4:e118.

4.       Methodology section is pretty good

5.       Result and discussion section, I suggest some latest references and make a valuable story.

I am happy to review the revised version.

Author Response

Thank you very much for your good comments and careful review. The responds to your comments are as following:

1.       Abstract looks good but just wondering if it can cover the whole theme

Response: The biochemical compositions were added in the abstract.

2.       The introduction should be better organized. Some of the sentences are not well structured, should be clarified and rewritten. It is advice to link the story in a better way in an introduction to convey a proper message to readers.

Response: We have revised this section according to these comments as shown in the introduction.

3.       I suggested some of the latest references, it would improve content and the novelty of the research. Authors can incorporate the related research carried out at the University of Queensland, Australia.

Marine-based species comprising approximately one half of the total universal biodiversity, the oceans and aquatic environment in general offer a plenty of resource for novel bioactive components. Marine species comprises bioactive compounds and much attention has been paid to them as they play pivotal role in disease prevention and maintenance of human health. These marine bioactive compounds exhibit significant biological properties that contribute to their nutraceutical and pharmaceutical potential and are also considered to be safer alternatives to some existing synthetic drugs. These bioactivities include anti-oxidant, anti-thrombotic, anti-coagulant, anti-inflammatory, anti-proliferative, anti-hypertensive, anti-diabetic and cardio-protection activities, making them attractive nutraceuticals and pharmaceutical compounds.

·         Marine-Based Nutraceuticals: An Innovative Trend in the Food and Supplement Industries. Marine drugs, 13(10), 6336.

·         Current and potential uses of bioactive molecules from marine processing waste. J. Sci. Food Agric., 96: 1064–1067.

·         Therapeutic potential of abalone and status of bioactive molecules: A comprehensive review. Critical reviews in food science and nutrition, 57(8), 1742-1748.

·         Marine bioactive compounds and health promoting perspectives; innovation pathways for drug discovery. Trends in Food Science & Technology, 50, 44-55

·         Marine Processing Waste - In Search of Bioactive Molecules. Nat Prod Chem Res 4:e118.

Response: These contents have been added in the introduction (L28-35)

4.       Methodology section is pretty good

Response: We have further made some revisions.

5.       Result and discussion section, I suggest some latest references and make a valuable story.

Response: We have added some latest references, such as Felician et al (2018), Tabakaeva et al (2018), Bu et al (2017), Astre et al (2018), Luo et al (2018). We have made many revisions including addition of bioactive collagen peptides to make a valuable story.

Special thanks to you for your good comments and careful review.